# Research on the Three-Machines Perception System and Information Fusion Technology for Intelligent Work Faces

**DOI:** 10.3390/s23187956

**Published:** 2023-09-18

**Authors:** Haotian Feng, Xinqiu Fang, Ningning Chen, Yang Song, Minfu Liang, Gang Wu, Xinyuan Zhang

**Affiliations:** 1School of Mines, China University of Mining and Technology, Xuzhou 221116, China; 2Research Center of Intelligent Mining, China University of Mining and Technology, Xuzhou 221116, China; 3The 707th Research Institute of China State Shipbuilding Corporation Limited, Tianjin 300131, China; 4Tianjin Navigation Instruments Research Institute, Tianjin 300131, China

**Keywords:** intelligent mining, intelligent working face, attitude monitoring, cooperative control, perceptual information fusion, optical fiber sensing, three-machines linkage

## Abstract

The foundation of intelligent collaborative control of a shearer, scraper conveyor, and hydraulic support (three-machines) is to achieve the precise perception of the status of the three-machines and the full integration of information between the equipment. In order to solve the problems of information isolation and non-flow, independence between equipment, and weak cooperation of three-machines due to an insufficient fusion of perception data, a fusion method of the equipment’s state perception system on the intelligent working surface was proposed. Firstly, an intelligent perception system for the state of the three-machines in the working face was established based on fiber optic sensing technology and inertial navigation technology. Then, the datum coordinate system is created on the working surface to uniformly describe the status of the three-machines and the spatial position relationship between the three-machines is established using a scraper conveyor as a bridge so that the three-machines become a mutually restricted and collaborative equipment system. Finally, an indoor test was carried out to verify the relational model of the spatial position of the three-machines. The results indicate that the intelligent working face three-machines perception system based on fiber optic sensing technology and inertial navigation technology can achieve the fusion of monitoring data and unified expression of equipment status. The research results provide an important reference for building an intelligent perception, intelligent decision-making, and automatic execution system for coal mines.

## 1. Introduction

Since the release of the Guiding Opinions on Accelerating the Development of Intelligent Coal Mines, the coal industry has vigorously promoted the construction of intelligent coal mines with technological innovation as its core driving force. The proposal of the national double carbon strategy further promotes the construction of a multi-industry chain and multi-system integrated intelligent coal mining system in the coal industry, moving towards a modern mining path of green intelligence, safety, efficiency, and sustainable development [1,2]. The process of intelligent construction in coal mines generates a large amount of information and data, and new technological means are used for data fusion and analysis, which is in line with the national strategy of “building a digital China” and “adhering to innovation driven development” [3].

With the advancement of intelligent construction in coal mines, coal mine production has achieved the goal of reducing personnel in the working face and is moving towards the goal of fewer or even unmanned personnel [4,5]. The hydraulic support, scraper conveyor, and shearer (three-machines), as the core equipment of fully mechanized mining faces, have been equipped with basic attitude and position perception functions, and new technologies are constantly emerging and applied to the field of coal mining to help improve the state perception level of the three-machines.

Due to the harsh environment and technological limitations in coal mines, traditional methods, such as gear counting, ultrasonic reflection, and infrared reflection, are used to monitor coal mining machines, all of which have many problems [6]. Fang Xinqiu et al. [7] used the inertial navigation system to monitor the shearer, but also faced the problem of low accuracy. Fan Qigao et al. [8] created an error analysis model using the extended Kalman filter to improve the positioning accuracy of strapdown inertial navigation system (SINS) shearer. Yang Hai et al. [9] used a motion constraint-assisted SINS zero velocity update model to improve the positioning accuracy during the operation process. Chen Yuming et al. [10] proposed an initial alignment compensation strategy based on the mechanical model of coal mining machinery to solve the performance degradation problem of the underground initial alignment algorithm in coal mines. WU Gang et al. [11,12] proposed an SNIS error compensation model through theoretical analysis and simulation experiments and conducted experimental verification. Ge Shirong et al. [13] developed a device for detecting the position and attitude of the shearer based on the working face geographic information system with inertial navigation device and shaft encoder as sensing elements. Zhang Boyuan et al. [14] proposed two deviation angle calibration algorithms based on the two-point method. 

The attitude detection objects of hydraulic support mainly include the support height and the attitude angle of base, top beam, and cover beam. Vaze Jai et al. [15] established a mathematical model to calculate the posture in a hydraulic support using the length and angle of the column and connecting rod. Zhang Yi et al. [16] proposed a straightness monitoring method for advanced hydraulic supports in order to monitor the position and posture of the supports in the roadway. Jiao Xiubo et al. [17] proposed an intelligent decision-making method for the hydraulic support position and attitude self-adjustment based on digital twin system driving. Chen Ningning et al. [18,19] developed an inclination sensor and pressure sensor based on fiber Bragg grating technology and built the attitude detection method of hydraulic support based on fiber sensing technology. Chen Hongyue et al. [20] proposed a measurement method for the attitude angle of the top beam of advanced hydraulic supports based on visual inspection technology. Zeng Qingliang et al. [21] proposed a binocular vision detection method to solve the problem of difficult real-time pose and straightness detection of hydraulic supports in unmanned coal mine working faces under complex geological conditions. 

The detection methods of scraper conveyor straightness can be divided into the following two categories: indirect detection and direct detection. The indirect method is mainly to invert the shape of the scraper conveyor through the shearer or hydraulic support, while the direct method is to directly detect the shape of the scraper conveyor body. In 2003, the Australian Federal Academy of Sciences (CSIRO) applied a strapdown inertial navigation technology to automated coal mining technology [22]. Kelly et al. [23] obtained the position and attitude information of the shearer based on inertial navigation technology, which can indirectly reverse the straightness state of the scraper conveyor. Fang Xinqiu et al. [24,25,26,27] developed a three-dimensional curvature (shape) sensor based on fiber optic sensing technology for the real-time dynamic perception of the motion form of scraper conveyors, and developed a curve reconstruction algorithm to achieve three-dimensional shape detection of scraper conveyors. Hao Shangqing et al. [28] constructed a geometric measurement model for the scraper conveyor based on the position of the shearer. Song Yang et al. [29] constructed a precision compensation model for the rotational error angle to improve the accuracy of fiber Bragg grating (FBG) curvature sensors for scraper conveyors. Xie Jiacheng et al. [30] used a collaborative analysis solution method to calculate the posture of a curved scraper conveyor. Li Suhua et al. [31] proposed a straightening method based on the motion law of a floating connection mechanism and established a virtual straightening system for “coal seams and equipment”.

Through the continuous efforts of researchers, the state perception system of the three-machines is becoming more and more perfect. However, the perception system of the three-machines is independent of each other, and the data are not intercommunicating, resulting in the processing degree of the perception information being unable to support the linkage control of the three-machines, which seriously hinders the development of the perception data fusion decision and the collaborative control of the intelligent three-machines. In this paper, an intelligent sensing system of three-machines on the working face based on optical fiber sensing and inertial navigation technology is constructed, and a fusion method of the sensing information of the three-machines is further proposed. The information of the three-machines is fused in the same reference frame, and the location information of the three-machines is transferred between the devices. This research is of great significance for the coordination and intelligent control of the three-machines.

## 2. Intelligent Perception System for Three-Machines Status

### 2.1. Optical Fiber Sensor

Fiber optic sensing technology has the advantages of high accuracy and strong anti-interference ability. After reasonable packaging, it can achieve strong environmental tolerance, and the networking methods are flexible and diverse [32,33]. After proposing the unmanned working face mining technology system, Fang et al. [34] conducted research on the application of fiber optic sensing technology in coal mining, such as the basic principles of fiber optic sensing technology, information sensing principles, mechanical models of information transmission, packaging technology, etc. 

In order to cope with the complexity of coal mine application scenarios, the author’s team proposed the following three FBG packaging technologies: surface adhesive, grooved landfill, and surface adhesive substrate, and further developed the FBG pressure sensor [19], FBG inclination sensor [20], and FBG 3D curvature sensor [31]. It can not only ensure accuracy, but also work stably in the complex coal mining environment for a long time [35], providing a set of reliable sensors for the construction of intelligent coal mining face. 

### 2.2. Scraper Conveyor Straightness Fiber Sensing System

The spatial shape of the scraper conveyor is affected by the complex floor conditions and the cumulative error of the hydraulic support pushing action; moreover, its body shape presents different degrees of bending in the horizontal and vertical directions. The bending section is too long, which increases the time for the shearer to cut one cut of coal. The bending section is too short, and pushing the scraper conveyor chute may cause the scraper chain to top [36]. The straightness perception of the scraper conveyor is helpful to adjust the straightness of the scraper conveyor body in advance to avoid stopping production due to bending of the body. The FBG 3D curvature sensor and spatial curve reconstruction algorithm were used to build an intelligent sensing system for the straightness of the scraper conveyor using FBG (Figure 1), which is used to obtain real-time online 3D bending shape information of the scraper conveyor [25]. The FBG 3D curvature sensor is shown in Figure 2. 

According to coal mine safety regulations, the temperature of the working face should be below 26 °C. Moreover, the working surface temperature of modern intelligent mines is at the optimal temperature suitable for the human body, which is considered a constant temperature environment, so the impact of temperature on FBG is ignored.

### 2.3. Hydraulic Support State Fiber Sensing System

The hydraulic support not only supports the top plate of the working face to provide a safe coal cutting space for the shearer, but also drives the scraper conveyor forward to achieve a continuous and safe advancement of the mining face by completing the four basic actions of lifting, lowering, pushing, and moving [37]. If the hydraulic support exhibits abnormal attitudes, such as tilting, collision, or tilting of the top beam, it will seriously affect the advancement of the working face and the safety of the mining space; therefore, hydraulic support state perception is an important part of intelligent coal mining face construction. The FBG inclination sensor and FBG pressure sensor are used to build a fiber intelligent sensing system for hydraulic support. Based on the hydraulic support attitude perception index and the kinematics model, real-time online perception system of the hydraulic support base and link attitude, working resistance, and top beam attitude is realized (Figure 1). The FBG pressure sensor and the FBG inclination sensor are shown in Figure 3. 

### 2.4. Shearer State Perception System

The position and attitude detection of shearer is an important foundation for achieving intelligent production in fully mechanized mining faces. The spatial walking trajectory of the shearer not only reflects the straightness of the scraper conveyor, but also affects the timeliness of the hydraulic support movement execution. As shown in Figure 4, an advanced inertial navigation device is installed on the body of shearer to obtain the position and attitude information during its operation. However, the inertial measurement unit (IMU) has poor adaptability in complex environments, such as underground coal mines, making it difficult to achieve accuracy when working on the ground. Therefore, Wu Gang et al. [8,9] proposed a shearer attitude error compensation model based on fiber optic strapdown inertial navigation to improve the perception accuracy of IMU in coal mines and obtain real-time and accurate information about the position and operating attitude of the shearer. 

## 3. Three-Machines State Perception Information Fusion

### 3.1. Establishment of Datum Coordinate System

The attitude and spatial position relationship information of the three-machines is the basis for determining their constraint relationships, working parameters, task coordination strategies, etc. In the linkage relationship of the three-machines, the scraper conveyor is like a bridge connecting the coal mining machine and the hydraulic support, and at the same time, the three also form a special relationship body that is both collaborative and mutually constrained. The relational model of the spatial position of the three-machines will be proposed in this part, which can assist the three-machines intelligent control system to carry out equipment information perception, data transmission, data fusion analysis, and intelligent decision-making. However, in fact, the on-site operators in the coal mine can easily and quickly locate the equipment through the serial number of the hydraulic support, so when building the relational model of the spatial position of the three-machines, the serial number information of the hydraulic support is introduced, which is convenient for on-site operators to use. 

The state information of the three-machines on the intelligent working face is fused and described in the same coordinate system, which can easily and quickly express and calculate the spatial position, attitude, and relative spatial relationship of the three-machines. The coordinate systems often used to describe the state of the equipment include the geocentric inertial coordinate system, earth coordinate system, geographic coordinate system, navigation coordinate system, and body coordinate system. Therefore, an appropriate reference system should be selected as the datum coordinate system of the working face, and then the local coordinate system should be determined according to the operation characteristics of the equipment. The state information of the equipment is transformed from the local coordinate system to the common reference coordinate system to complete the fusion expression of the attitude and position information of the equipment. In this paper, a datum coordinate system (d-system) is established with the midpoint of the scraper conveyor head as the origin, the positive direction of Z axis points to the sky, and the positive direction of X axis points to the advancing direction of the working face. The positive direction of the Y axis is parallel to the working face and conforms to the right-hand rule. When the scraper conveyor is in an ideal straight state, the Y axis coincides with the central axis of the scraper conveyor body, as shown in Figure 4. 

### 3.2. Scraper Conveyor Bend Section Description

Assuming that the body of the scraper conveyor is connected by *n* + 1 middle chutes through n connection points. The FBG sensing units is installed at n positions in the FBG 3D curvature sensor, corresponding to the connection of the middle chute of the scraper conveyor, and can obtain real-time position curvature information. The information of these n positions can be obtained by the interpolation and reconstruction algorithm of the FBG sensing system of the scraper conveyor straightness, and they are described as *P* in the reference coordinate system (d system) as follows: (1)$$P=\left\{p_{1},p_{2},p_{3},\cdots ,p_{n}\right\}=\left\{\left(x_{1},y_{1},z_{1}),(x_{2},y_{2},z_{2}),\cdots ,(x_{n},y_{n},z_{n}\right)\right\}$$

Due to the fact that the scraper conveyor is composed of multiple middle chutes with a length of *s*, when the body is in a bent state, the middle chute will not deform, while the adjacent middle chutes will move together to form a certain angle. Therefore, the coordinates of the middle chute connection point will be used to describe the bending section of the scraper conveyor body.

As shown in Figure 5, first sort the middle chutes (1, 2, 3... *n* + 1) of the scraper conveyor and the FBG (1, 2, 3... *n*) in the FBG 3D curve sensor. Suppose that the scraper conveyor starts to bend from the middle chute of section j and section k in the horizontal and vertical directions, respectively, and the middle chute of section (*x* + 1) is bent in both directions, then the curvature of the bending section in the horizontal and vertical directions are expressed as $\overset{⏜}{q_{j-1}q_{j+x}}$ and $\overset{⏜}{q_{k-1}q_{k+x}}$, and the coordinate points set *C*_1_ and *C*_2_ of (*x* + 2) the FBG positions corresponding to the bending section and the coordinates of the beginning and end points of the bending section are, respectively, as follows: (2)$$\left\{\begin{array}{l} C_{1}=\left\{q_{j-1},q_{j},q_{j+1},\cdots ,q_{j+x}\right\}\\ q_{j-1}=(x_{j-1},y_{j-1},z_{j-1})\\ q_{j+x}=(x_{j+x},y_{j+x},z_{j+x}) \end{array}\right.,\left\{\begin{array}{l} C_{2}=\left\{q_{k-1},q_{j},q_{k+1},\cdots ,q_{k+x}\right\}\\ q_{k-1}=(x_{k-1},y_{k-1},z_{k-1})\\ q_{k+x}=(x_{k+x},y_{k+x},z_{k+x}) \end{array}\right.$$

The angle at which the scraper conveyor deviates from the position of the fuselage under ideal conditions in both the horizontal and vertical directions is used to represent the bending degree of the fuselage (Figure 6). The angle of deviation is represented by the angle between the *Y* axis and the line, and the line is the maximum deviation point and departure point in the bending section. The deviation angle *φ_cH_* of horizontal bending $\overset{⏜}{q_{j-1}q_{j+x}}$ and the deviation angle *γ_cV_* of vertical bending $\overset{⏜}{q_{k-1}q_{k+x}}$ are calculated as follows: (3)$$\left\{\begin{array}{l} \Delta x_{\max H}=\max\left\{\left|x_{j+1}-x_{j}\right|,\left|x_{j+2}-x_{j}\right|,\cdots ,\left|x_{j+x}-x_{j}\right|\right\}\\ \Delta y_{\max H}=\max\left\{\left|y_{j+1}-y_{j}\right|,\left|y_{j+2}-y_{j}\right|,\cdots ,\left|y_{j+x}-y_{j}\right|\right\}\\ \varphi_{cH}=\arctan(\Delta x_{\max H}/\Delta y_{\max H}) \end{array}\right.$$
(4)$$\left\{\begin{array}{l} \Delta y_{\max V}=\max\left\{\left|y_{k+1}-y_{k}\right|,\left|y_{k+2}-y_{k}\right|,\cdots ,\left|y_{k+x}-y_{k}\right|\right\}\\ \Delta z_{\max V}=\max\left\{\left|z_{k+1}-z_{k}\right|,\left|z_{k+2}-z_{k}\right|,\cdots ,\left|z_{k+x}-z_{k}\right|\right\}\\ \gamma_{cV}=\arctan(\Delta z_{\max V}/\Delta y_{\max V}) \end{array}\right.$$
where, $\Delta x_{\max H}$, $\Delta y_{\max H}$, $\Delta y_{\max V}$ and $\Delta y_{\max V}$ denote the length of the bending section in the *XYZ* direction, respectively. 

### 3.3. Hydraulic Support Status Description

#### 3.3.1. Hydraulic Support Attitude Description

In Figure 7, local coordinate systems *O_ia_X_ia_Y_ia_Z_ia_* (*O_ia_*-system), *O_ib_X_ib_Y_ib_Z_ib_* (*O_ib_*-system) and *O_ic_X_ic_Y_ic_Z_ic_ (O_ic_*-system) are used to describe the attitude of the base, cover beam, and top beam of the hydraulic support, respectively, where *i* denotes the support number, a, b, and c denote the base, cover beam, and top beam, re spectively. And the initial state of the local coordinate system is consistent with the reference coordinate system. The coordinate system conversion is used to describe the attitude of each part of the hydraulic support in the d-system. The conversion of d-system to the local coordinate system is achieved by rotating *φ, θ, γ* around *Z*, *X,* and *Y,* respectively, three times as follows: $$OXYZ\overset{Z}{\underset{C_{\varphi }}{\to }}O_{1}X_{1}Y_{1}Z_{1}\overset{X}{\underset{C_{\theta }}{\to }}O_{2}X_{2}Y_{2}Z_{2}\overset{Y}{\underset{C_{\gamma }}{\to }}O_{i}X_{i}Y_{i}Z_{i},$$

Therefore, the solution equation describing the attitude of the three parts of the hydraulic support is as follows:(5)$$(C_{d}^{ia},C_{d}^{ib},C_{d}^{ic})=C_{\gamma }C_{\theta }C_{\varphi }$$
(6)$$C_{\varphi }=\left(\begin{array}{ccc} \cos\varphi & -\sin\varphi & 0\\ \sin\varphi & \cos\varphi & 0\\ 0 & 0 & 1 \end{array}\right),\;C_{\theta }=\left(\begin{array}{ccc} 1 & 0 & 0\\ 0 & \cos\theta & \sin\theta \\ 0 & -\sin\theta & \cos\theta \end{array}\right),\;C_{\gamma }=\left(\begin{array}{ccc} \cos\gamma & 0 & -\sin\gamma \\ 0 & 1 & 0\\ \sin\gamma & 0 & \cos\gamma \end{array}\right)$$
(7)$$\begin{array}{l} C_{\gamma }C_{\theta }C_{\varphi }=\left(\begin{array}{ccc} \cos\gamma & 0 & -\sin\gamma \\ 0 & 1 & 0\\ \sin\gamma & 0 & \cos\gamma \end{array}\right)\left(\begin{array}{ccc} 1 & 0 & 0\\ 0 & \cos\theta & \sin\theta \\ 0 & -\sin\theta & \cos\theta \end{array}\right)\left(\begin{array}{ccc} \cos\varphi & -\sin\varphi & 0\\ \sin\varphi & \cos\varphi & 0\\ 0 & 0 & 1 \end{array}\right)\\ =\left(\begin{array}{ccc} \cos\gamma \cos\varphi +\sin\gamma \sin\theta \sin\varphi & \sin\varphi \cos\theta & \sin\gamma \cos\varphi -\cos\gamma \sin\varphi \sin\theta \\ -\cos\gamma \sin\varphi +\sin\gamma \sin\theta \cos\varphi & \cos\varphi \cos\theta & -\sin\gamma \sin\varphi -\cos\gamma \sin\varphi \cos\theta \\ -\sin\gamma \cos\theta & \sin\theta & \cos\gamma \cos\theta \end{array}\right) \end{array}$$

#### 3.3.2. Hydraulic Support Position Description

The origin *O_ia_* of the local coordinate system of the hydraulic support base is selected as a reference point to describe the position of the hydraulic support (Figure 7). The vector $\vec{OO_{ia}}$ denotes the position of a hydraulic support in the d-system at some time. Therefore, the vector denotes the position of the hydraulic support in the d-system at this time as follows: (8)$$\vec{OO_{ia}}=\left(x_{ia},y_{ia},z_{ia}\right)=\left(OA,OB,O_{ia}{O_{ia}}^{\prime }\right)$$
where, *OA* denotes the distance from the origin *O_ia_* of the local coordinate system *O_i_*_a_-system to the *Y* axis of the d-system; *OB* denotes the distance from the origin *O_ib_* of the local coordinate system *O_ia_*-system to the *X* axis of the d-system; *O_ia_O_ia_*′ denotes the distance from the origin of *O_ia_*-system *O_ia_* to the plane of *X_ia_O_ia_Y_ia_*, and *O_ia_*′ denotes the projection point of *O_ia_* on the d-system *XOY* plane. When the bottom plate of the working face is in an ideal horizontal state, the value of *O_ia_O_ia_*′ is 0. 

#### 3.3.3. Analyze the Position Coordinates of the Hydraulic Support

The hydraulic supports are arranged in the order of 1, 2, 3… *i* from the vicinity of the machine head to the tail (Figure 7). The arrangement relationship between the hydraulic supports and the middle chute of the scraper conveyor is shown in Figure 8, and the positions of n FBGs are all located in the middle of the two hydraulic supports. The number of hydraulic supports in the intelligent working face is more than the number of FBG positions, so there is a quantitative relationship between the serial number of FBGs and the serial number of hydraulic supports. It is only necessary to determine the serial number of hydraulic supports on both sides of the position of the No.1 FBG when arranging the FBG 3D curvature sensor. Assuming that the No.1 FBG position is located between the No.*N* and No.(*N* + 1) hydraulic support, and the No.*j* FBG position is located between the No.*i* and No.(*i* + 1) hydraulic support, the relationship between *i* and *j* is as follows: (9)i=j+N−1

The No.*i* hydraulic support is located between the No.*j* and No.(*j*−1) FBG positions. For any hydraulic support numbered *i*, the local coordinate system *O_ia_X_ia_Y_ia_Z_ia_* of its base and the spatial relationship between the hydraulic support and its corresponding central slot are shown in Figure 8. Point *O_ia_* (*x_ia_*, *y_ia_*, *z_ia_*) can be calculated according to point *O_j_* (*x_j_*, *y_j_*, *z_j_*) and coordinate point *O_j_*−1 through square (10).
(10)$$\left\{\begin{array}{l} x_{ia}=x_{j}-d_{x}-d_{p}=x_{j-1}+d_{x}-d_{p}=x_{j-1}-d_{p}+(x_{j}-x_{j-1})/2\\ y_{ia}=y_{j}-d_{y}=y_{j-1}+d_{y}=y_{j-1}+(y_{j}-y_{j-1})/2\\ z_{ia}=z_{j}=z_{j-1} \end{array}\right.$$
where, *dx* and *d_y_* denote the distance between the two adjacent FBG positions in the *X* and *Y* axis directions, respectively, and *d_p_* denotes the distance between the base origin *O_ia_* and the middle chute, which takes a value according to the state of the hydraulic support. When the hydraulic support is pushed, *d_p_* takes a maximum value *d_pmax_*. After the following maneuvering is completed, *d_p_* is reduced to *d_pmin_*. 

Equation (10) can be used to represent the spatial position relationship between the hydraulic support and the scraper conveyor, while retaining the relationship between the scraper conveyor and the serial number of the hydraulic support. The position of the bending section of the scraper conveyor can be represented by the serial number of the hydraulic support. 

### 3.4. Shearer’s Condition Description

#### 3.4.1. Shearer Attitude Description

The fiber optic inertial measurement unit used to obtain the attitude information of the shearer also needs to transform coordinate system (Figure 9). Similar to the description of the attitude of the hydraulic support, it describes the attitude information of the shearer in d-system. As with Equation (4), the conversion of the same d-system to the *O_b_X_b_Y_b_Z_b_* (*O_b_*-system) of the shearer is realized by rotating *φ*, *θ*, and *γ* around *Z*, *X*, and *Y,* respectively, by three times of a single coordinate axis. The conversion of coordinate system can be realized by combining Equations (11) and (6).
(11)$$C_{d}^{b}=C_{\gamma }C_{\theta }C_{\varphi }$$

#### 3.4.2. Shearer Position Description

When the shearer moves back and forth along the scraper conveyor on the working face, the origin of the shearer’s *O_b_*-system is used as a reference point to describe the spatial position of the shearer. Under ideal conditions, the running trajectory of the shearer is flat and coincides with the central axis of the scraper conveyor body, which means that the trajectory of its straight section is parallel to the *Y* axis of the d-system. In fact, the running trajectory of shearer is not a straight line, but fluctuates around the *Y* axis of the d-system as the body of the scraper conveyor bends. Assuming that at a certain moment, the shearer runs to the position shown in Figure 9, and the vector $\vec{OO_{b}}$ represents the position of the shearer in the d-system at this time: (12)$$\vec{OO_{b}}=\left(x_{b},y_{b},z_{b}\right)=\left(OA,OB,OC\right)$$
where, *OC* = *O_b_*′*O_b_*, *O_b_*′ denotes the projection of *O_b_* on the *XOY* plane. Equation (12) only shows the vector expression of the shearer position, and the coordinates of the vector need to be obtained through the inertial navigation system installed on the shearer. The specific process is not described here. This section will study the mathematical relationship between the serial numbers of the shearer and the hydraulic support, based on the obtained position coordinate *O_b_* ($x_{b},y_{b},z_{b}$) of the shearer. 

The characteristic of the movement of the shearer is to perform a back-and-forth motion on the scraper conveyor; that is, to perform a back-and-forth motion along the *Y* axis of the d-system. Therefore, determining the relationship between the shearer and the hydraulic support only requires considering the Y value of the coordinate points of the two positions. As shown in Figure 10, the spatial relationship between the b-system origin *O_b_* of the shearer and the origin *O_ia_* of a certain hydraulic support base at a certain moment is shown. The position of point *O_b_* is mostly between two adjacent hydraulic supports. Equation (13) is used to roughly calculate the serial numbers of the hydraulic supports corresponding to the shearer. Assuming that there are also *k* hydraulic supports in the negative *Y* axis direction of the d-system, the calculation result of Equation (13) is taken as an integer upwards, and then Equation (14) is used for serial number calibration. The calibration range is about k hydraulic supports.
(13)$$i=\left\lceil y_{b}/d_{h}\right\rceil +1$$
where, *y_b_* denotes the *Y* coordinate value of the shearer position. *D_h_* denotes he distance between hydraulic supports. ⌈ ⌉ denotes the integer symbol taken upwards.
(14)$$△y_{s}=\min\left\{\left|y_{ia}-y_{b}\right|,\left|y_{(i+1)a}-y_{b}\right|,\left|y_{(i+2)a}-y_{b}\right|,\left|y_{(i+3)a}-y_{b}\right|,\left|y_{(i+4)a}-y_{b}\right|\right\}$$
where, Δ*y_s_* denotes the actual distance value between the hydraulic support and the shearer in the *Y* axis direction. *Y*_(*i*+2)*a*_ denotes the *Y* axis coordinate value of the No.(*i* + 2) hydraulic support, (*i* + 2) denotes the serial number of the hydraulic support. A denotes the local coordinate system of the hydraulic support base. 

At the minimum value of *y_s_*, the corresponding hydraulic support serial number is the serial number of the hydraulic support closest to the position of the shearer.

## 4. Three-Machines Spatial Position Relational Model Verification Test

An indoor test system was set up to verify the mathematical model of the spatial relationship between the three machines. The indoor test platform was built with the straightness sensing system of the scraper conveyor as the center. As shown in Figure 11, the FBG 3D curvature sensors containing 7 FBG positions were laid, and the FBG positions were ranked from 1 to 7 from right to left. Then, the 8 middle chutes and 10 hydraulic supports are simulated on both sides of the FBG 3D curvature sensor. The length of the middle chutes is *l_c_* = 1.75 m and the width *b* = 0.8 m, and the sequence is from 1 to 8. The distance between the hydraulic support *d_h_* = 1.75 m, the distance between the hydraulic support and the middle chute *d_p_* = 1.2 m, in order of from 1 to 10, where the negative direction of the *Y* axis hydraulic support one (*k* = 1); that is, the position of the first FBG is between the second and the third hydraulic support (*N* = 1). Then, Equation (9) is transformed into the following: *i* = *j* + 1. 

The spatial position relationship of the three-machines is mainly related to the *X* and *Y* axis directions, so this model validation test only considers the case of the scraper conveyor body in the *XOY* plane.

### 4.1. The Scraper Conveyor Is in a Straight Line State 

As shown in Figure 11, the following is a set of coordinate points for 7 FBG positions: *C_s_*_1_ = {*q*_1_, *q*_2_*, q*_3_…*q*_8_} = {(−0.45, 1.77, 0), (−0.44, 3.53, 0), (−0.44, 5.29, 0), (−0.45, 7.05, 0), (−0.45, 8.80, 0), (−0.45, 10.44, 0), (−0.46, 12.20, 0)}. 

The following set of coordinate points of the hydraulic support can be calculated using Equation (10): *C_h_*_1_ = {*h*_1_, *h*_2_, *h*_3_…*h*_10_} = {(−1.65, −0.88, 0), (−1.65, 0.88, 0), (−1.65, 2.65, 0), (−1.64, 4.41, 0), (−1.65, 6.17, 0), (−1.65, 7.92, 0), (−1.65, 9.62, 0), (−1.65, 11.32, 0), (−1.65, 13.13, 0), (−1.65, 14.88, 0)}.

The following set of three shearer position coordinate points is randomly given: *C_sh_*_1_ = {*s*_1_, *s*_2_, *s*_3_} = {(−0.52, 4.23, 0), (−0.56, 7.45, 0), (−1.53, 10.76, 0)}. 

According to Equations (13) and (14), the corresponding hydraulic numbers of the shearer are {4, 6, 8}. 

### 4.2. The Scraper Conveyor Is in a Local Bending State

As shown in Figure 12, the following is set of coordinate points for 7 FBG positions: *C_s_*_2_ = {*q*_1_, *q*_2_*, q*_3_…*q*_8_} = {(−0.45, 1.77, 0), (−0.45, 3.46, 0), (−0.45, 5.24, 0), (−0.56, 7.03, 0), (−0.59, 8.56, 0), (−0.97, 10.00, 0), (−2.17, 11.78, 0)}. 

The following set of coordinate points of the hydraulic support can be calculated using Equation (10): *C_h_*_2_ = {*h*_1_, *h*_2_, *h*_3_…*h*_10_} {(−1.67, −0.88, 0), (−1.67, 0.88, 0), (−1.67, 2.61, 0), (−1.70, 4.36, 0), (−1.73, 6.14, 0), (−1.76, 7.79, 0), (−1.97, 9.29, 0), (−2.77, 10.89, 0), (−2.77, 12.64, 0), (−2.77, 14.39, 0)}. 

The following set of three shearer position coordinate points is randomly given: *C_sh2_* = {*s*_1_, *s*_2_, *s*_3_} = {(−0.45, 6.46, 0), (−0.45, 9.86, 0), (−0.45, 11.50, 0)}.

According to Equations (13) and (14), the corresponding hydraulic numbers of the shearer are {5, 7, 8}.

The test results show that the spatial position relationship model of the three-machines can realize the fusion and flow of the position relationship of the three-machines, and the position coordinates of the hydraulic support can be obtained according to the shape information of the scraper conveyor, which is of great significance for the linkage and cooperative control of the three-machines on the working face.

## 5. Conclusions

(1)The intelligent perception system of the three-machines in the intelligent working face has been successfully built based on fiber optic sensing technology and inertial navigation technology, and the fusion expression of the state perception information of the three-machines has been achieved. (2)The datum coordinate system of the working face was established at the head position of the scraper conveyor and successfully used to describe the position and attitude information of the three-machines equipment.(3)The spatial position relationship model of the three-machines in this article has been established and successfully applied to the fusion of position information of the three-machines, and its feasibility has been verified through indoor experiments.(4)Although we have verified the feasibility of the fusion method, we have not conducted system integration, so we have not yet compared it with existing systems, and the required reaction time and latency of the fusion method have not been verified. This article also did not consider the impact of temperature changes, so temperature compensation design should be added in environments with large temperature differences. These contents will become the next research focus.

## Figures and Tables

**Figure 1 sensors-23-07956-f001:**
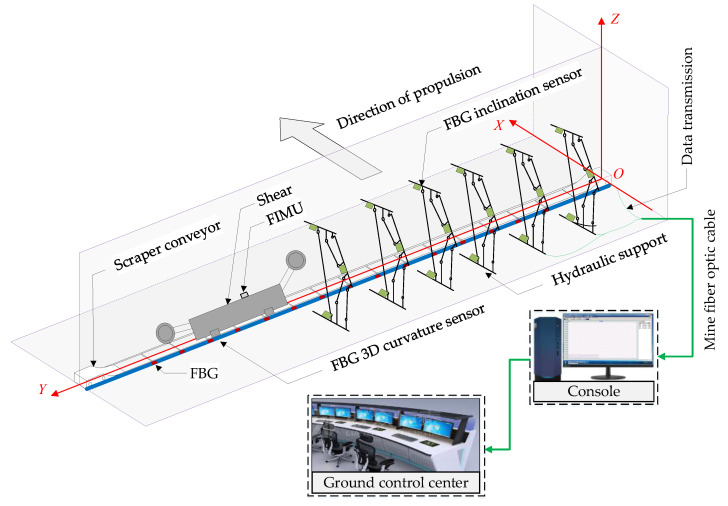
Establishment of d-system and three-machines state sensing system for working face.

**Figure 2 sensors-23-07956-f002:**
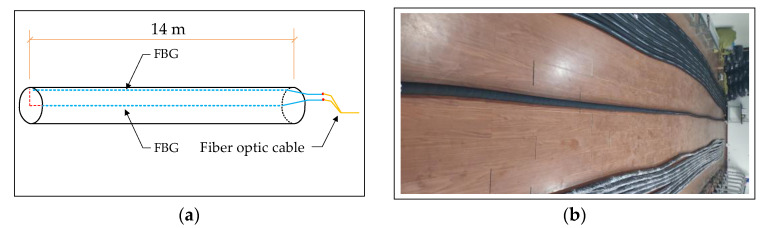
Schematic diagram and product diagram of FBG 3D curvature sensor (**a**) structural schematic diagram of FBG 3D curvature sensor; (**b**) FBG 3D curvature sensor products.

**Figure 3 sensors-23-07956-f003:**
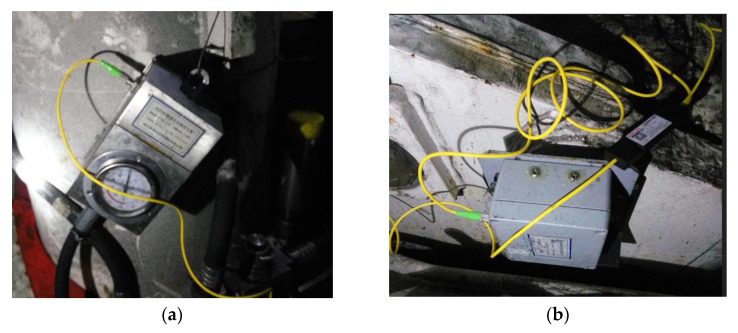
Hydraulic support status sensing sensor. (**a**) FBG inclination sensor; (**b**) FBG pressure sensor.

**Figure 4 sensors-23-07956-f004:**
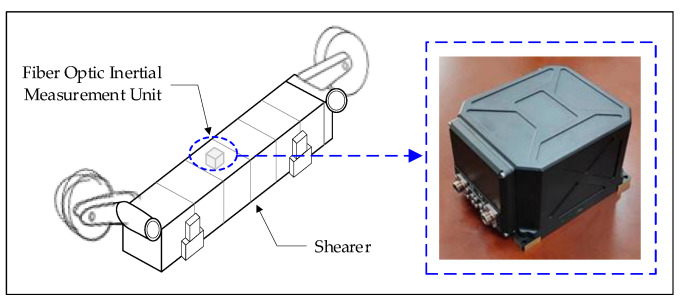
Position and attitude perception of shearer based on IMU.

**Figure 5 sensors-23-07956-f005:**
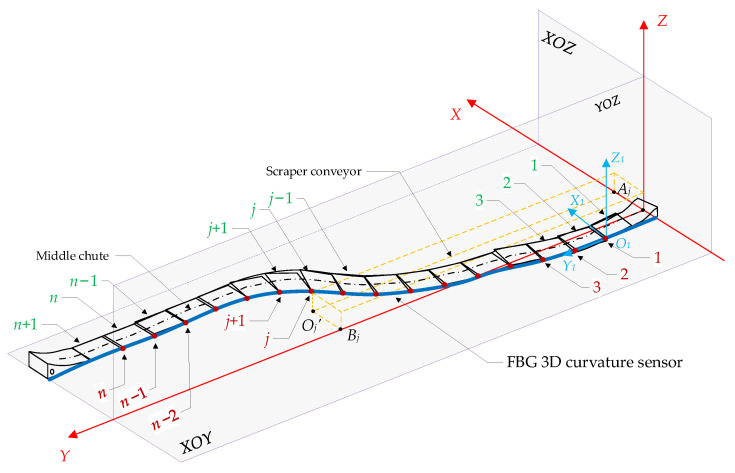
Layout diagram of straightness perception system for scraper conveyor.

**Figure 6 sensors-23-07956-f006:**
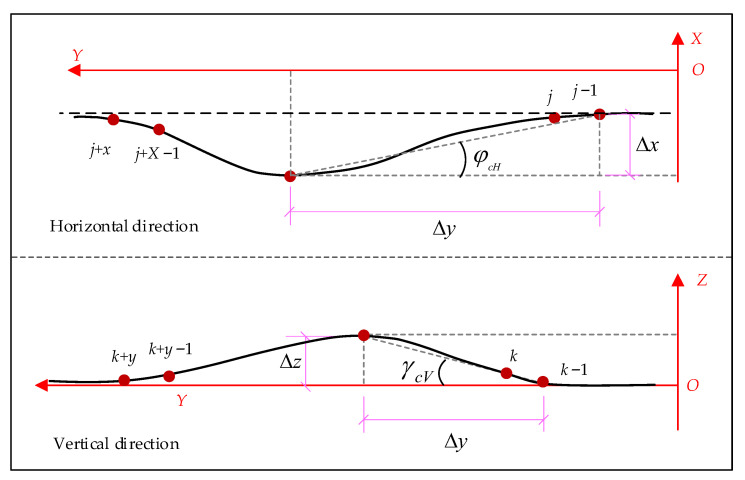
Schematic diagram of bending section of scraper conveyor.

**Figure 7 sensors-23-07956-f007:**
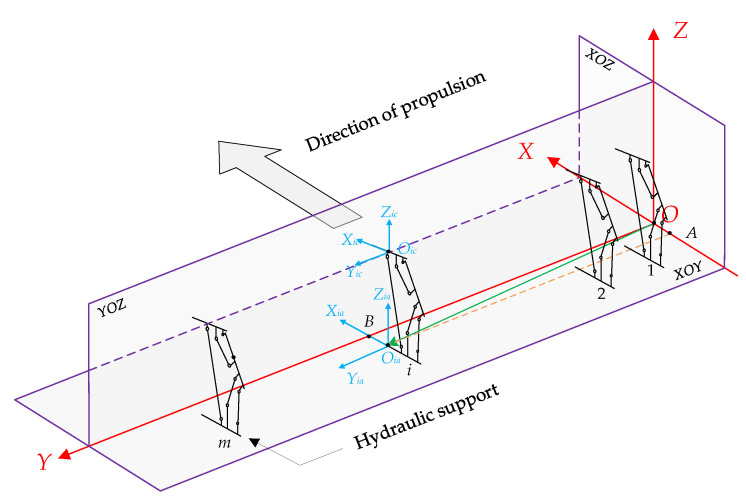
Schematic diagram of position and attitude perception system for hydraulic support.

**Figure 8 sensors-23-07956-f008:**
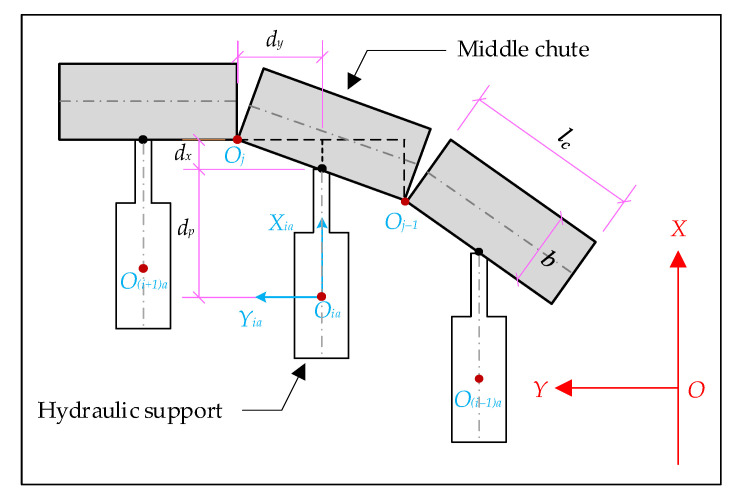
Schematic diagram of the position relationship between hydraulic support and FBG.

**Figure 9 sensors-23-07956-f009:**
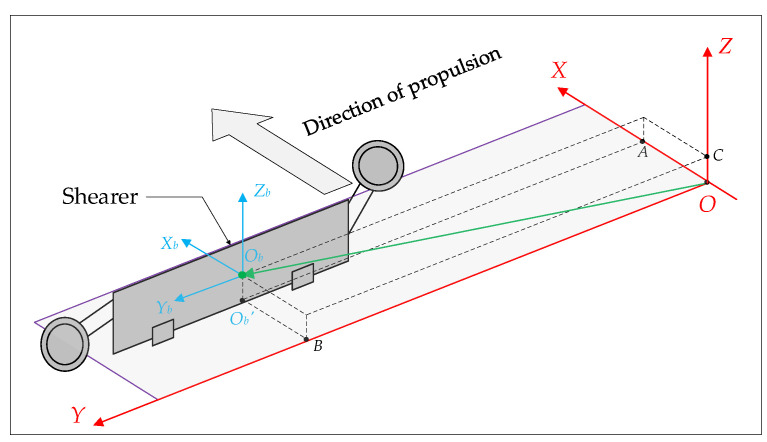
Description of shearer status information.

**Figure 10 sensors-23-07956-f010:**
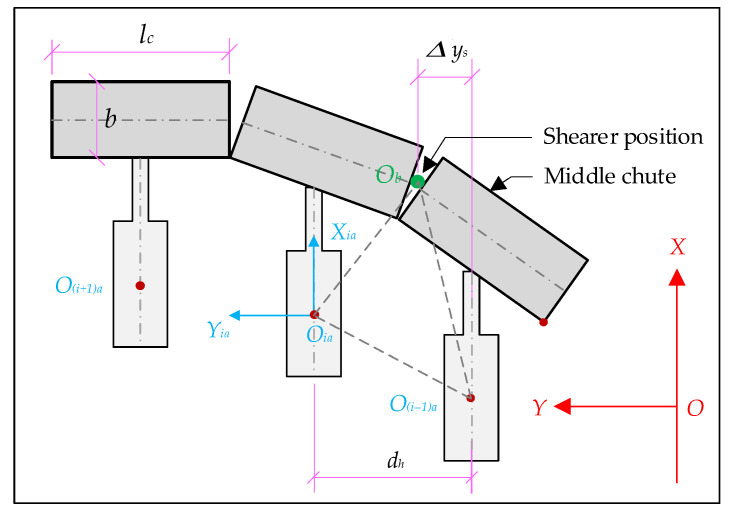
Diagram of the position relationship between shearer and hydraulic support.

**Figure 11 sensors-23-07956-f011:**
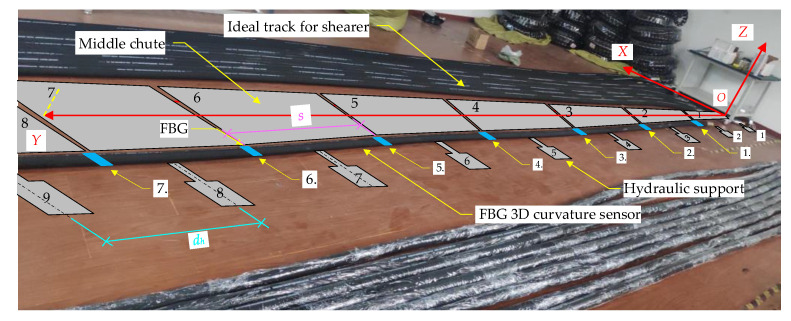
Scraper conveyor under straight line conditions.

**Figure 12 sensors-23-07956-f012:**
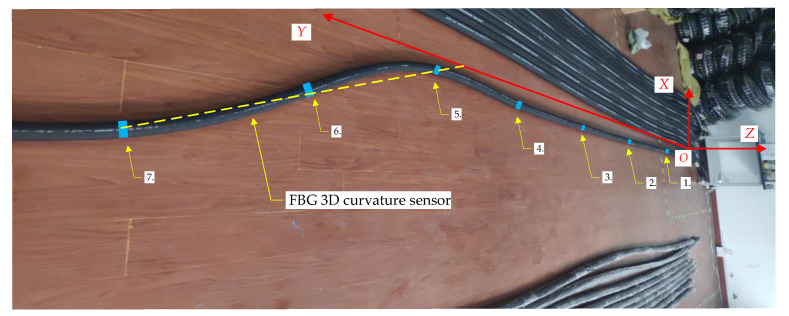
Scraper conveyor under bending conditions.

## Data Availability

All data and code used or analyzed in this study are available from the corresponding author on reasonable request.

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
