# Peer review of "Research on the Three-Machines Perception System and Information Fusion Technology for Intelligent Work Faces"

_sensors, 2023, doi:10.3390/s23187956_

Round 1

Reviewer 1 Report

The manuscript of sensors-2593835 is to propose the three-machines perception system and information fusion technology for intelligent work faces. The manuscript may be interesting and can be accepted in Sensors after minor revision.

1.Intelligent work faces includes shearer, scraper conveyor and hydraulic support. Is there a time requirement for information fusion among the shearer, scraper conveyor and hydraulic support, and what is the reaction time?

2. The three-machines perception system and information fusion technology for intelligent work faces proposed in the article has not been compared with other systems.

There are some writing and gramma mistakes in this paper, the author should check the.

Reviewer 2 Report

The manuscript under review presents an intelligent sensor system for coal mining equipment. In general, the manuscript has a reasonable structure and clearly presents the results. However, several issues must be addressed before the article can be published.

1) A paragraph after equation 1 beginning with “Due to the fact that the scraper…” is repeated twice.

2) Please, correct the caption of Figure 7.

3) A fiber Bragg grating (FBG) is a key component in the proposed 3D curvature sensors. However, an FBG can only detect deformation in one direction (typically, stretching or compression along its axis, which can also result from bending). In this regard, it is not clear how the three-dimensional curvature is measured using FBGs. Please provide additional information about the sensor structure and the positions of FBGs inside it, and briefly describe the method of 3D bending detection based on the data from FBGs.

4) There is also lack of information about the FBG interrogation device(s) used in the proposed system, such as its manufacturer/model, wavelength range, number of channels, resolution, accuracy, sampling rate, etc. It is also recommended to establish requirements (minimal acceptable parameters) for such devices in the proposed application.

5) In the case when FBG sensors are multiplexed using wavelength division method, it is necessary to estimate the possible maximum shift of the FBG Bragg wavelengths in order to eliminate the superposition of their spectra. Please, provide information on the maximum possible FBG strain and the resulting Bragg wavelength shift of the sensors in the 3D curvature sensors.

6) In the real-life applications of the FBG-based sensor systems it is crucial to take into account the ambient temperature impact on the FBG wavelength change. How is it achieved in the proposed system?

Round 2

Reviewer 2 Report

The authors have responded to all the concerns.